# Emotional Exhaustion, Depersonalization, and Mental Health in Nurses from Huelva: A Cross-Cutting Study during the SARS-CoV-2 Pandemic

**DOI:** 10.3390/ijerph18157860

**Published:** 2021-07-25

**Authors:** Francisco-Javier Gago-Valiente, María-Isabel Mendoza-Sierra, Emilia Moreno-Sánchez, Félix Arbinaga, Adrián Segura-Camacho

**Affiliations:** 1Health Department of IES Cuenca Minera, Minas de Riotinto, 21660 Huelva, Spain; 2Department of Social, Development and Educational Psychology, Faculty of Education, Psychology and Sports Sciences, University of Huelva, 21007 Huelva, Spain; imendoza@dpsi.uhu.es (M.-I.M.-S.); adrian.segura@dpee.uhu.es (A.S.-C.); 3Department of Pedagogy, Faculty of Education, Psychology and Sports Sciences, University of Huelva, 21007 Huelva, Spain; emilia@dedu.uhu.es; 4Department of Clinical and Experimental Psychology, Faculty of Education, Psychology and Sports Sciences, University of Huelva, 21007 Huelva, Spain; felix.arbinaga@dpsi.uhu.es

**Keywords:** nursing staff, pandemics, coronavirus infections, professional burnout, mental health

## Abstract

Currently, healthcare professionals are particularly vulnerable to the impact of the SARS-CoV-2pandemic since they directly deal with patients suffering from this disease and are in the first line of fire, which increases their risk of contagion. This research examines the prevalence of emotional exhaustion, depersonalization, and possible non-psychotic psychiatric disorders in 48 male and 270 female nursing professionals of Huelva during the COVID-19 pandemic. To this end, we analyzed the relationship between these dependent variables and considered various sociodemographic variables. The nursing staff of public hospitals in Huelva who have had contact with cases of SARS-CoV-2 in their work environment showed a poorer state of mental health than that of others of this same professional category who have not had contact with this type of situation.

## 1. Introduction

Healthcare professionals are frequently confronted with life and death situations, so their work can become physically and emotionally exhausting, with fatigue, low self-esteem, depersonalization, and professional isolation being the main problems faced by such workers [1].

In addition to these worrying issues, the health emergency generated by the pandemic known as COVID-19, produced by the coronavirus (SARS-CoV-2), has aggravated the psychological discomfort experienced by working professionals, with healthcare personnel being an unavoidable reference [2,3].

Adverse psychological reactions have already been reported among healthcare workers during other epidemics such as the SARS outbreak [4,5], MERS [6], and currently SARS-CoV-2 [7]. These studies found that these workers feared contagion and infection of their family members, friends, and colleagues [6], experiencing feelings of uncertainty and stigmatization [4,5], leading to long-term psychological consequences [8]. Moreover, as shown in other epidemics, stress and anxiety have a direct impact on the health of staff, which indirectly affects the healthcare system because of health professionals taking sick leave [9].

Currently, healthcare professionals are susceptible to this problem, since they directly deal with patients suffering from this disease and are on the front line, increasing their risk of contagion.

Some reports have recently been published on the effects of SARS-CoV-2 on the mental health of healthcare staff [10]. However, despite the findings concerning other pandemics, there is still a lack of sufficient information on the most devastating pandemic of the last century, that is, COVID-19. Therefore, this study was designed to determine the prevalence of emotional exhaustion, depersonalization, and possible non-psychotic psychiatric disorders in nursing professionals working in the public health system of the province of Huelva (Andalusia, Spain) during the current pandemic state of emergency. In particular, the present work analyzed the relationship between these dependent variables and considered a number of sociodemographic variables.

It should be noted that although the aim of this research is not to detect burnout syndrome, two of the variables that we intend to analyze are common symptoms of it. Traditionally, this syndrome has been detailed in the scientific literature in a three-dimensional way, formed by emotional exhaustion, depersonalization, and low personal fulfillment. However, a large amount of research has shown that emotional exhaustion and depersonalization are the main aspects of the syndrome, considering personal fulfillment as an independent role related to a more stable aspect of the personality. Emotional exhaustion and depersonalization are two inseparable parts of the syndrome. Therefore, this study has focused only on these two aspects [11,12,13,14].

## 2. Materials and Methods

### 2.1. Participants

This was a descriptive cross-cutting study in which 318 nursing professionals participated, of whom 15.1% were men between the ages of 27 and 63 years (n = 48), and 84.9% were women between the ages of 22 and 64 years (n = 270).The distribution of the sample according to work center was as follows: 63.2% (n = 201) were from the Juan Ramón Jiménez Hospital complex, and 36.8% (n = 117) from the Infanta Elena Hospital. Regarding marital status, 46.7% were married, 30.9% were single, 4.8% were divorced, 3% were widowed, and 14.6% were in a relationship. Of the participants, 64.1% had children and 35.9% did not.

### 2.2. Procedure

The participants were recruited from two public hospitals of Huelva Capital: the Juan Ramón Jiménez Hospital Complex and the Infanta Elena Hospital. Data collection was carried out in April, May, and June 2020. A probabilistic model was used for sample selection, considering a sampling error of 0.05. A work schedule was drawn up, and visits to the hospitals were organized in the morning and afternoon shifts and across different units and services to avoid duplicating information. During each visit, paper questionnaires were given to each professional for 30 min in the presence of the researcher or collaborator in case clarifications were needed. All participants provided written informed consent to analyze and publish their data, and their anonymity was assured. The response rate was 79.7% for nurses at the Juan Ramón Jiménez Hospital Complex and 81.2% for those at the Infanta Elena Hospital. Among the reasons given for not responding to the surveys was that they had too much work or lack of time. The management of both hospitals and the University of Huelva Research Ethics Committee authorized the present study with the registration code TD-EPSH-2019 and the internal code 1585-N-19.

Figure 1 highlights the evolution of the number of SARS-CoV-2 cases confirmed by PDIA (Active Infection Diagnostic Tests) in the districts of the province of Huelva, diagnosed during fieldwork period:

### 2.3. Evaluation Instruments

The participants completed a brief sociodemographic questionnaire providing information on their age, gender, work center, marital status, and parental status. They had to provide their email along with sociodemographic data in order to be able to identify possible duplicates. At the end of this questionnaire, an item was introduced to ask the study population if they had been in contact with any SARS-CoV-2 cases in their work environment.

The symptoms of emotional exhaustion and depersonalization were evaluated through the Maslach Burnout Inventory-Human Services Survey (MBI-HSS) questionnaire. This questionnaire was chosen due to its reliability indices of 0.90 for emotional exhaustion and 0.79 for depersonalization, and internal consistency of 0.80 for all items [16]. Other studies [17] using the MBI have reported a Cronbach’s alpha coefficient of 0.78 for the emotional exhaustion aspect and 0.71 for the depersonalization aspect. Thus, the MBI can be considered appropriate for this research. This questionnaire is self-administered and is designed to evaluate the feelings and attitudes of the professional towards their work and patients. It consists of 22 items, and measures (among other variables) emotional exhaustion and depersonalization. The cut-off point for identifying people with high emotional exhaustion was ≥27 points in the sum of the items corresponding to this aspect within the scale and ≥10 points for high depersonalization [18].

Potential non-psychotic psychiatric disorders were assessed using the 12-item short form of the General Health Questionnaire (GHQ-12). This instrument is designed to detect non-psychotic psychiatric pathologies in the general population [19]. The results of Spanish validation studies and the recommendations of the authors of the questionnaire used a cut-off point of ≥12 to identify people who may have mental or emotional disorders [20,21]. The GHQ-12 has been validated in Spain and has been widely used to assess the general population [19,21,22]. Other researchers [23] conducted validation studies of the questionnaire in a sample of 1641 participants, obtaining adequate internal consistency and showing overall Cronbach’s alpha coefficient results of 0.90. Finally, studies conducted in other countries, such as that of Burrone et al. [24], have also shown good psychometric properties and reliability of the instrument in a population of 854 people, with a Cronbach’s alpha of 0.80.

### 2.4. Statistical Analysis

All statistical analyses were conducted with the SPSS (Statistical Products and Service Solutions) version 23.0 statistical software.

First, it should be noted that a univariate analysis was carried out, which included the calculation of the mean, standard deviation, maximum value, and minimum value of the variables of age, emotional exhaustion (quantitative data), and depersonalization (quantitative data).

We also calculated frequencies and percentages for the variables of gender, marital status, paternity/maternity, work center, emotional exhaustion, depersonalization, contact with SARS-CoV-2, and self-perceived general health (likely or unlikely non-psychotic psychiatric case).

Normality tests of the quantitative variables were used to determine whether we could apply parametric or nonparametric tests in subsequent analyses. In addition, since the number of data points used was greater than 50, the Kolmogorov–Smirnov statistic was selected for the normality tests.

Subsequently, statistical tests were conducted according to the objective of this study. Details of the analyses are described below:

Cross-tabulations were created to compare emotional exhaustion and depersonalization as a function of gender, work center, marital status, and parenthood. In addition, chi-square tests were conducted between these variables.

Because the normality tests for the variables of age, emotional exhaustion (quantitative), and depersonalization (quantitative) revealed an abnormal distribution, the Mann–Whitney U test was used for tests of independence with two categorical variables, and the Kruskal–Wallis H test was used when there were three or more groups. Correlations between the different study variables were also analyzed using Spearman’s rho.

Cross-tabulations were created regarding the possible presence of non-psychotic psychiatric disorder according to emotional exhaustion and depersonalization and contact with SARS-CoV-2. In addition, Chi-square tests were conducted between these variables.

Cross-tabulations were created to compare GHQ12 scores according to gender, marital status, and parental status. In addition, chi-square tests were also conducted between these variables.

## 3. Results

### 3.1. Differences in the Variables of Emotional Exhaustion and Depersonalization According to Gender

The variables of emotional exhaustion and depersonalization differed according to gender ((χ^2^ = 10.320; *p*= 0.006) and (χ^2^ = 58.728; *p* = 0.000), respectively, at a 95% confidence level). It was evidenced that women presented a lower percentage of affectation in these variables compared to men (Figure 2).

### 3.2. Prevalence of Emotional Exhaustion and Depersonalization as a Function of the Work Center

It should be noted that the Juan Ramón Jiménez Hospital Complex presented a higher percentage of people with high emotional exhaustion (44.6% versus 38.4%), while a higher percentage of participants from the Infanta Elena Hospital showed depersonalization (35% versus 27.4%).

### 3.3. Differences in Emotional Exhaustion and Depersonalization According as a Function of Age, Marital Status, and Parental Status

Significant differences were found in emotional exhaustion and depersonalization according to age (Kruskal–Wallis H = 28.557 and 22.959, respectively, and *p* = 0.000 at a 95% confidence level in both dimensions). The mean age of participants with high emotional exhaustion was 47.4 years compared with mean age of 42.2 years for those with low emotional exhaustion. For participants with high depersonalization, the mean age was 47.9 years compared with 44.9 years for those with medium–-low depersonalization.

When studying the correlation coefficient between the variables of emotional exhaustion and depersonalization with the age variable, a positive but very low correlation was evidenced (Table 1).

Regarding the possible relationship between marital status and emotional exhaustion and depersonalization, the greatest percentages of people with high emotional exhaustion and high depersonalization were those who have a partner and are married (53.7% and 33.7%), respectively. In contrast, the lowest scores on both dimensions were obtained by widowed and divorced people (0% and23.5%, respectively) (Table 2).

No differences were found in emotional exhaustion and depersonalization according to parental status ((χ^2^ = 4.360; *p* = 0.113) and (χ^2^ = 4.154; *p* = 0.125), respectively, at a 95% confidence level).

### 3.4. Relationship between the Variables of Emotional Exhaustion and Depersonalization and Contact with SARS-CoV-2 in the Work Environment and GHQ-12 Score

A greater percentage of people who had contact with SARS-CoV-2 in their work environment showed high levels of emotional exhaustion (49.6%) and depersonalization (34.3%) than people who had no contact (38.3% and 21.1%, respectively).

Among the cases of emotional exhaustion, there were around 60% with non-psychotic psychiatric symptoms compared to 28.5% who did not show it. On the other hand, inthe cases of depersonalization, almost 40% evidenced non-psychotic psychiatric symptoms, compared to 25% who did not (Figure 3).

### 3.5. Relationship between the Possible Presence of Non-Psychotic Psychiatric Pathology (Positive GHQ-12) and Gender, Age, Marital Status, and Parental Status

A higher percentage of men obtained a positive GHQ-12 score than that of women (Table 3).

Regarding marital status, the highest percentage of positive scores on the GHQ-12 was found among widows, while the lowest percentage of positive scores was obtained by those with a partner. In addition, the *p*-value of the chi-square test was significant (*p* = 0.000 at a 95% confidence level), so both variables have dependence in the selected sample (Table 3).

No statistically significant differences were found in GHQ-12 scores according to parental status. However, a positive GHQ-12 score was obtained by a higher percentage of people without children than people with children (Table 3).

Significant differences were found (Mann–Whitney U = 115915.500; *p* = 0.041 at a 95% confidence level) in GHQ-12 scores according to age. Those who obtained a positive score to indicate probable non-psychotic psychiatric pathologies had a mean age of 45.4 years, while those who obtained a negative score had a higher mean age of 46.7 years.

### 3.6. Relationship between Probable Non-Psychotic Psychiatric Disorder and Contact with SARS-CoV-2 in the Work Environment

The results revealed that a higher percentage of positive GHQ-12 scores was found among those individuals who had been in contact with SARS-CoV-2(58.7%) than those who had not (41.3%). Moreover, the *p*-value of the chi-square test of independence was significant (χ^2^ = 62.483; *p* = 0.000 at a 95% confidence level), indicating that the presence of possible non-psychotic psychiatric disorders differs as a function of SARS-CoV-2 contact.

## 4. Discussion

The conclusions of other studies [25,26,27,28] agree with many of the results reported in the present investigation. Significant gender differences were found in emotional exhaustion and depersonalization, with women showing lower levels of suffering than men. Other studies [26] attribute and relate high emotional exhaustion to low personal fulfillment, with men being more affected than women. These gender differences could be based on the link between gender and other characteristics and sociodemographic variables. A certain gender could be associated with the greater presence of other variables that act as modulators or accentuators of emotional exhaustion and depersonalization [29].

The greater incidence of high emotional exhaustion among the staff of the Juan Ramón Jiménez Hospital Complex could be explained by the fact that, although both centers offer the same services, the palliative care service and the chronic care unit of the Juan Ramón Jiménez Hospital Complex include a large number of patients and professionals. Thus, in agreement with the findings of other studies [27], it can be concluded that these professionals experience greater emotional exhaustion due to the more significant number of services involving factors such as pain or death. Other studies [30] have reported a relationship between low personal fulfillment and depersonalization, which was observed in the personnel of the Hospital Infanta Elena, where more professionals are affected by emotional exhaustion.

The results of numerous investigations exploring the link between age, emotional exhaustion, and depersonalization have yielded discrepant results, as the relationship established between age and these variables has been direct in some studies, such as Cabrera and Elvira [31], and less so for others, such as Carlotto [32]. Nonetheless, it should be noted that the results concerning age should always be interpreted with caution due to the problem of survivor bias. That is, it is likely that those who suffer from early career burnout will leave their jobs, leaving behind those who survive and who, consequently, present lower levels of burnout [33].

On the other hand, some studies [34] show higher levels of emotional exhaustion or depersonalization in married people or those with a partner, while others have found these levels to be higher in single people [33]. According to the results obtained in this research, it appears that analyzing marital status as the only influence of family on work could involve several biases. Family support does not necessarily only come from the partner, as a person can receive support from a father, mother, niece, nephew, child, cousin, or other relative.

Regarding parental status, although it could be thought that being a parent negatively impacts the professional career of an individual and, therefore, their fulfillment due to lack of time [35], no such findings were obtained in the population studied here. It would also be interesting to analyze whether workers feel satisfied with the family reconciliation measures applied in each center.

On the other hand, there may be people who, although showing high emotional exhaustion, high depersonalization, and poor self-perceived general health, manifest high personal fulfillment because the achievement of their career goals allows them to overcome challenges at work [36]. This could cause this symptomatology to go unnoticed by the affected individuals.

In contrast to other studies [37], in the present study, no significant gender differences were found in GHQ-12 scores (*p* = 0.076 at a 95% confidence level). This finding could be related to other data from this study showing the lower percentages of women with positive scores on other mental health indicators, such as the MBI-HSS.

Another study conducted on populations of caregiving professionals suggests that a professional’s perception of their health improves as the irage of increases, although no statistically significant differences were found [28]. This pattern of results also emerged in the sample of the present study.

The results of other studies on healthcare personnel are in line with the results obtained in the present study, showing more mental health problems in people who do not have a partner compared with those who do [33].

Given the observed GHQ-12 scores according to marital status and parental status, it appears that the family environment could be acting as a modulating variable in the mental health of professionals, so that this symptomatology goes unnoticed in people who present these affectations [29,38].

The relationship found between depersonalization and a positive result in GHQ-12 could provide relevant data to take into account. Depersonalization can be a symptom of major depression. It has been shown that in patients suffering from unipolar depression, the associated depersonalization symptoms are more intense compared to healthy controls, and that there is also a positive correlation between depression and depersonalization [39]. These conditions are probably not discrete categories but have common biological basis and may be at least part of a continuum or spectrum of affective disorders [40].

Few studies have explored the relationship between mental or emotional problems of healthcare professionals in Spain with SARS-CoV-2since this is a very recent situation that is still ongoing. However, studies that have already investigated this issue have reported findings that are in line with the results presented in this paper. The data reveal that a high percentage of health professionals who have had contact with the pandemic tend to suffer symptoms of anxiety, stress, depression, sleep disorders, or other types of psychological problems [3,41,42,43,44].

Emotional exhaustion and depersonalization already presented a trend of greater affectation in men than in women before the pandemic [45]. However, the percentages of the population affected by both variables have been higher in this study carried out during the pandemic period [46,47]. It would be opportune to carry out mitigation strategies in pandemic situations. Since resources can be particularly scarce during these states, timely psychological support could also take many forms, including telemedicine and informal support groups [48].

Despite the richness and impact of the data obtained, the study described here has certain limitations and strengths. This is a cross-sectional study in which the relationship between variables is measured at a specific moment in time without a follow-up. Thus, many of the variables measured here could change, which, as we know, is happening due to the changes that have occurred in Spanish public healthcare [49]. This could modify the scores of many objective and subjective health indicators. In addition, it would also be appropriate to inquire about the psychiatric background of the participating population and include items in the data collection questionnaires on access to resources (social, economic, health, etc.) during the pandemic. This pandemic has left a mark on low socioeconomic status [50]. Therefore, it would be very interesting to carry out prospective studies over the same study sample, including new items with more variables, to re-assess the situation and potential changes. Another limitation concerns the sample size. Although the number of health professionals represents the reality of this group in the hospitals studied, we cannot extrapolate the results to other hospitals and provinces. Further studies will need to be carried out with larger sample sizes so that the results have greater external validity, and it will be necessary to recruit more health centers in which the professionals are classified according to the different services and units to which they belong. It would also be convenient to carry out correlation studies between the GHQ-12 outcome and emotional exhaustion and depersonalization. One of the strengths of this study is that it provides novel information on a series of mental health indicators in the healthcare population that have, until now, been rarely studied in the context of a pandemic situation. This updated information could help to inform future lines of intervention that address the problems studied here. Another important contribution of this study concerns the evidence regarding variations in mental health according to gender.

## 5. Conclusions

In summary, it can be concluded that the nursing staff of the public hospitals of Huelva capital who have had contact with SARS-CoV-2 in their work environment have a poorer state of mental health than that of those professionals who have not had contact with this situation. This state has been evidenced by high emotional exhaustion, high depersonalization, and a positive score to indicate the likely presence of non-psychotic psychiatric pathologies, with the latter constructs also showing a significant relationship with each other. Finally, it is worth mentioning that the men in this study sample showed, in general, a poorer state of mental health than that of the women.

## Figures and Tables

**Figure 1 ijerph-18-07860-f001:**
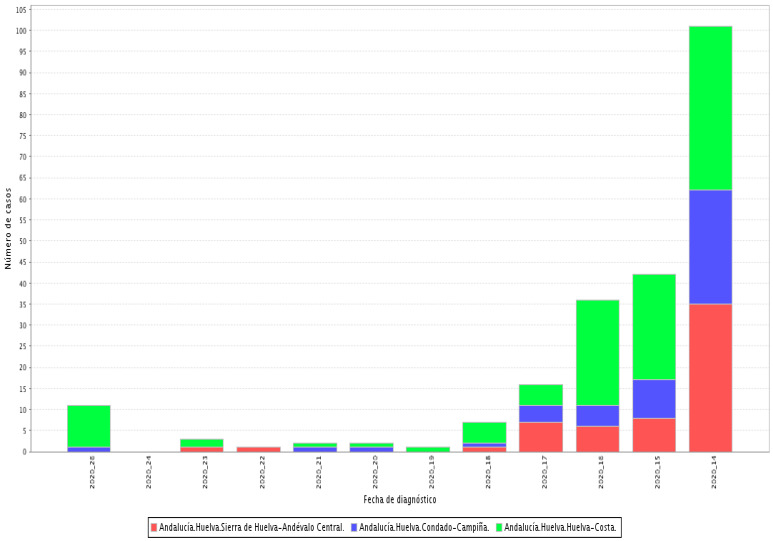
SARS-CoV-2 cases in the districts of the province of Huelva grouped by weeks [15].

**Figure 2 ijerph-18-07860-f002:**
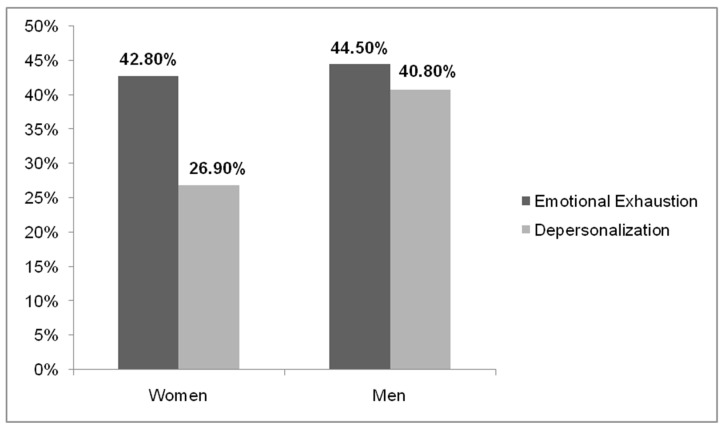
Percentages of emotional exhaustion and depersonalization in women and men.

**Figure 3 ijerph-18-07860-f003:**
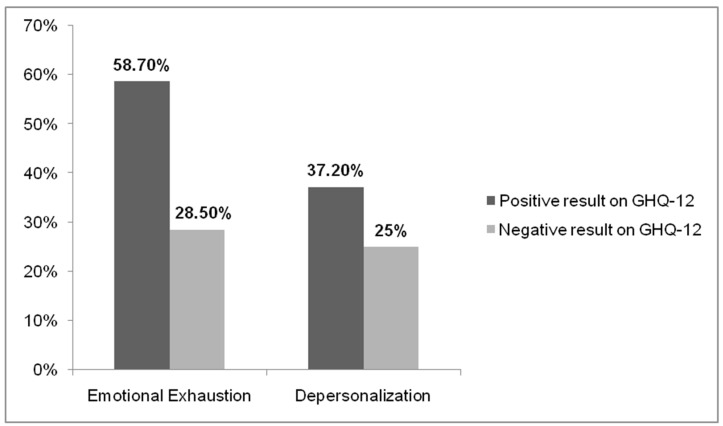
GHQ-12 results in people with emotional exhaustion and depersonalization.

**Table 1 ijerph-18-07860-t001:** Spearman’s rho correlation for the variables of emotional exhaustion and depersonalization with age.

Spearman’s Rho		Age	Emotional Exhaustion	Depersonalization
Age	Correlation coefficient	1.000	0.157 **	0.035
	Sig. (bilateral)	0.0	0.000	0.266
	N	318	318	318
Emotional exhaustion	Correlation coefficient	0.157 **	1.000	0.363 **
	Sig. (bilateral)	0.000	0.0	0.000
	N	318	318	318
Depersonalization	Correlation coefficient	0.035	0.363 **	1.000
	Sig. (bilateral)	0.266	0.000	0.0
	N	318	318	318

** The correlation is significant at the 0.01 level (bilateral).

**Table 2 ijerph-18-07860-t002:** Group statistics and Pearson’s chi-square test for marital status ^1^.

Variables	Emotional Exhaustion	Depersonalization
Marital status		
Pearson’s chi-square	67.688	60.043
Asymptotic significance (bilateral)	0.000 *	0.000 *

^1^ Grouping variables: emotional exhaustion and depersonalization. * *p*-value of the chi-square test.

**Table 3 ijerph-18-07860-t003:** Group statistics and Pearson’s chi-square test for the variables of sex, marital status, and people with children ^1^.

Independent Variables		Possible Presence of Non-Psychotic Psychiatric Pathology		Pearson’s Chi-Square	Asymptotic Significance (Bilateral)
		Yes (%)	No (%)		
Sex	Man	48.6%	51.4%	0.076	0.782 *
	Woman	47.6%	52.4%		
Marital status	Married	52.6%	47.4%	23.588	0.000 *
	Single	48.1%	51.9%		
	Divorced	39.2%	60.8%		
	Widower	64.3%	35.7%		
	With a partner	32.0%	68.0%		
People with children	Yes	46.2%	53.8%	1.909	0.167 *
	No	50.7%	49.3%		

^1^ Grouping variables: GHQ-12 results. * *p*-value of the chi-square test.

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
