# Peer review of "Emotional Exhaustion, Depersonalization, and Mental Health in Nurses from Huelva: A Cross-Cutting Study during the SARS-CoV-2 Pandemic"

_ijerph, 2021, doi:10.3390/ijerph18157860_

Round 1
Reviewer 1 Report
Thank you for this report on the mental health of nursing professionals in Huelva during these trying times. I have the following major and minor comments for the authors to consider.
Specific comments:
- Suggest to change the study title to make it more succinct and effective. It should also be stated that this was a cross-sectional study.
- Accuracy is essential, this is especially true when it relates to the actual topic of the paper. i.e. "SARS-CoV-2" is the name of the virus, while "COVID-19" is the name of the disease that SARS-CoV-2 can cause in people – they are not the same thing or interchangeable. "COVID-19" is also short for "Coronavirus Disease 2019" while "SARS-CoV-2" is short for "severe acute respiratory syndrome coronavirus 2".
- "Despite the findings concerning other pandemics, there is still a lack of sufficient information on the most devastating pandemic of the last century, that is, COVID-19" - there are in fact several published reports on the mental health effects of COVID-19. In the introduction section, suggest also to cite a recent narrative review on the impact of pandemic on the mental health of healthcare workers (citation: pubmed.ncbi.nlm.nih.gov/32603985).
- Some comments on the local COVID-19 situation on the ground during the period the survey was conducted would be helpful.
- As no personal identifiers were collected, the possibility of duplicate responses cannot be entirely excluded.
- Please change "27,245" to "27.245" and "0,001" to "0.001".
- "Significant differences were found (p ≤ 0.05)" - please provide the exact p values (with the associated 95% confidence intervals) instead of simply "p<0.05". Writing "p<0.05" alone is neither informative nor useful.
- Suggest to present some of the results in bar graph format to promote easy reading and visualization.
- "The conclusions of other studies [18]" - suggest to cite more than a single study here.
- Depersonalization could also be a symptom of major depression. It has been found that in patients suffering from unipolar depression, associated depersonalization symptomatology is more intense compared to healthy controls, and also that there is a positive correlation between depression and depersonalization (citation: pubmed.ncbi.nlm.nih.gov/15022041). These conditions are likely not discrete categories but rather, have common biological underpinnings and may form at least part of a continuum or affective disorder spectrum (citation: pubmed.ncbi.nlm.nih.gov/32557983).
- In the discussion section, as part of mitigation strategies, as resources could be particularly scarce during a serious pandemic situation, timely psychological support could also take many forms, including telemedicine and informal support groups (citation: pubmed.ncbi.nlm.nih.gov/32380875). This should be mentioned.
- Several other study limitations exist. The use of self-reported measures is also a study limitation. Did the authors adjust for baseline depression or past psychiatric history as a covariate? Additionally, socioeconomic status still varies over time in this age range. It is also clear that this pandemic has disproportionately impacted racial minorities and lower-income families (citation: pubmed.ncbi.nlm.nih.gov/32391864).
- There was no data availability statement. The underlying data should be made publicly available. If this was not possible, please provide a reason why.
Reviewer 2 Report
Thank you for doing this important work. This article aimed at analyse “Emotional Exhaustion, Depersonalization and Self-Perceived Health in Nursing Professionals in Huelva: A Study of Mental Health during the COVID-19 Pandemic”.
Title
The MBI refers to 3 concepts, emotional exhaustion, depersonalization and low personal accomplishment, where is this last concept?
Abstract
The authors indicate: “This research examines the prevalence of emotional exhaustion, depersonalization”, why the authors do not study “personal accomplishment”.
Methods
It was a correlational study or a cross-sectional study?
Evaluation instruments
Maslach Burnout Inventory-Human Services Survey (MBI-HSS) has 3 dimensions (emotional exhaustion, depersonalization and low personal accomplishment). The low personal accomplishment dimension is missing. The concept of burnout cannot be analysed by only 2 dimensions, since the concept has 3 dimensions
What were the cut-off points of the MBI?
Where is the correlation variables table?
Discussion
How have burnout levels changed from pre-pandemic research?
Round 2
Reviewer 1 Report
Thank you for the revisions.
Author Response
Thank you very much for the suggestions that made a better work